# Perceived Stress and Smoking Cessation: The Role of Smoking Urges

**DOI:** 10.3390/ijerph20021257

**Published:** 2023-01-10

**Authors:** María Barroso-Hurtado, Daniel Suárez-Castro, Carmela Martínez-Vispo, Elisardo Becoña, Ana López-Durán

**Affiliations:** Smoking and Addictive Disorders Unit, Department of Clinical Psychology and Psychobiology, University of Santiago de Compostela, 15782 Santiago de Compostela, Spain

**Keywords:** smoking urges, perceived stress, psychological treatment, abstinence

## Abstract

Despite the fact that perceived stress is related to abstinence smoking outcomes, no studies have investigated the mediational effect of specific tobacco-related variables on this relationship. This study aimed to explore the indirect effect of perceived stress on abstinence at the end of treatment through smoking urges. The sample comprised 260 treatment-seeking smokers (58.5% female; Mage = 46.00; SD = 11.1) who underwent psychological smoking cessation treatment. The brief version of the Questionnaire of Smoking Urges (QSU) and the Perceived Stress Scale (PSS14) were used. Mediation analyses were conducted in which smoking urges and their dimensions were potential mediators in the relationship between perceived stress and abstinence at the end of treatment. The results showed a non-significant direct effect of perceived stress on abstinence. However, a significant indirect effect was found through smoking urges (QSU-total) and, specifically, through smoking urges associated with the expectation of negative affect relief (QSU-Factor 2). A non-significant indirect effect through smoking urges related to the expectation of tobacco use as a pleasurable experience (QSU-Factor 1) was also found. Analyzing possible mediator variables could contribute to understanding previous conflicting data. These findings point to potential interest in including treatment components targeting perceived stress and smoking urges to improve the effectiveness of smoking cessation treatments.

## 1. Introduction

Tobacco smoking is the leading preventable cause of morbidity and mortality worldwide [1], being responsible for eight million deaths per year [2]. Smoking contributes to numerous physical illnesses, including cancer, cardiovascular and respiratory diseases [3], and mental health problems, such as depression, anxiety, and schizophrenia [4]. Research has shown that smoking cessation could reverse, at least in part, these negative tobacco-related effects [5,6,7]. For example, quitting smoking has been shown to be related to improvements in mental health [8]. According to Hajek et al. [9] and Streck et al. [10], abstinence is associated with lower levels of stress and anxiety. Additionally, Kim et al. [11] found that successful quitters had lower odds of feeling stress than those who failed to quit. Other studies have shown that quitting smoking is associated with improvements in depressive symptoms [12,13] and may even decrease suicide risk [14].

There are numerous effective smoking cessation treatments [15]. For instance, behavioral interventions targeting smoking cessation have been shown to improve quit rates at a 6-month follow-up compared to usual/standard care, no intervention, and less intensive or alternative interventions (e.g., interventions with equal or lower intensity, different providers, or different theoretical bases) [5]. Recent studies have identified certain psychological variables that influence the effectiveness of these smoking cessation treatments, such as depression [16], anxiety [17], and self-efficacy [18]. Given the importance of these factors, it is important to continue investigating which personal variables influence whether people succeed in quitting.

One personal variable associated with smoking behavior is perceived stress, defined as the thoughts or feelings a person has about the level of stress they are experiencing during a specific period [19]. The relationship between perceived stress and smoking-related variables has been analyzed in different populations [20,21,22,23,24]. However, limited research has examined the relationship between perceived stress and smoking cessation treatment outcomes, and the results are inconsistent. Previous studies have found that higher baseline perceived stress in adult smokers was a significant predictor of lower abstinence rates at a 1-year follow-up in women but not in men after receiving physician’s advice, self-help materials, nicotine replacement therapy, and telephone counseling [25]. Moreover, al’Absi et al. [26] found an association between higher perceived stress at baseline and higher relapse likelihood after participants received one screening and orientation session for quitting. However, other studies have not found a significant association between baseline perceived stress and smoking cessation outcomes [27,28]. Using a placebo treatment combined with a smoking cessation guide and brief counseling sessions, Manning et al. [28] found a non-significant association between perceived stress and abstinence at the end of treatment and at a 6-month follow-up. Similarly, Hooper and Kolar [27] found a non-significant association between perceived stress and abstinence at the end of treatment and at 3- and 6-month follow-ups after cognitive behavioral treatment combined with transdermal nicotine patches. Considering this variety of results, it is necessary to investigate this relationship further and examine which factors may have influenced these findings.

One possible variable that might help clarify the relationship between perceived stress and quitting smoking is smoking urges or cravings, defined as a subjective experience of the desire to consume tobacco [29,30]. This variable has been extensively studied in relation to smoking behavior. For instance, stronger smoking urges are associated with a greater likelihood and faster progression to smoking [31], greater nicotine dependence [32,33], quit attempt failures [34], lower abstinence rates at the end of smoking cessation treatment [35], and elevated relapse rates [36,37,38].

Tiffany and Drobes [29] proposed that smoking urge is a multidimensional construct that includes the desire to smoke due to the expectation of a pleasurable experience and the desire to smoke associated with the expectation of relief from negative affect through smoking. Few studies have analyzed the differential influence of smoking urge dimensions on smoking cessation outcomes, and their findings vary. For instance, Allen et al. [39] conducted a study with women and found that those with higher scores on both smoking urge dimensions on their quit day had a higher risk of relapse within 30 days after quitting. However, Toll et al. [40] found that greater decreases in the desire to smoke scores only associated with the expectation of tobacco use as a pleasurable experience predicted relapse. Non-significant effects were found in the smoking urge related to the expectation of negative affect relief. Finally, in a study conducted in an experimental setting, Heffner et al. [41] found that short-term abstinent smokers with alcohol dependence showed stronger cravings associated with the expectation of negative affect relief only compared to smokers without alcohol dependence. In summary, most existing studies examining the effects of smoking urge dimensions on treatment outcomes have focused on abstinence maintenance or relapse risk. Minimal research has been conducted to investigate the effect of these dimensions on abstinence achievement after treatment.

Regarding the relationship between perceived stress and the urge to smoke, some studies have found that smoking urgency has been positively associated with perceived stress [42]. However, few studies have examined the relationship between the two dimensions of smoking urges and perceived stress. For instance, Lawless et al. [22], who investigated the association between perceived stress and smoking behavior according to sex, found a significant and positive correlation between perceived stress and smoking urges related to negative affect relief expectations in males only and not in females.

To our knowledge, no studies have examined the mediational effect of smoking urges and their dimensions on the relationship between perceived stress and abstinence achievement. Therefore, the aim of this study was to explore the indirect effect of perceived stress on abstinence outcomes at the end of treatment through smoking urges and their dimensions (desire and intentions to smoke and desire to smoke associated with the expectation of relief from negative affect).

## 2. Materials and Methods

The study procedures were reviewed and approved by the Bioethics Committee of the University of Santiago de Compostela before data were collected. Participants provided written consent after the research procedures had been explained to them.

### 2.1. Participants

The sample was composed of 260 Spanish treatment-seeking smokers who received face-to-face cognitive behavioral treatment to quit smoking at the Smoking Cessation and Addictive Disorders Unit of the University of Santiago de Compostela. The inclusion criteria were: (1) aged 18 years or older; (2) wishing to voluntarily participate in the treatment; (3) smoking at least six cigarettes/day; (4) completing the entire pre-treatment assessment; and (5) providing written informed consent. The exclusion criteria were: (1) diagnosis of a severe mental disorder (e.g., schizophrenia or bipolar disorder); (2) concurrent dependence on other substances (e.g., alcohol or cannabis dependence); (3) having a condition that implied high life risk that required immediate intervention (e.g., chronic obstructive pulmonary disease or lung cancer); and (4) not attending the first treatment session.

### 2.2. Measures

The following instruments were used at pre-treatment:The Smoking Habit Questionnaire [43]. This instrument is composed of 59 items about different aspects related to smoking behavior. It measures sociodemographic variables (sex, age, marital status, and educational level), history of past and current physical illnesses, and variables related to tobacco use (e.g., number of cigarettes per day [CPD], previous quit attempts, reducing tobacco use, stages of change, and use of other tobacco products different from cigarettes).The Fagerström Test for Cigarette Dependence (FTCD) [44]. This self-report questionnaire contains six items with two to four response options. The cut-off point that indicates high nicotine dependence is a score ≥ 6 [45]. We used the Spanish adaptation of the scale, which has a Cronbach’s alpha coefficient of 0.66 [46]. In our sample, the Cronbach alpha was 0.59.The brief version of the Questionnaire of Smoking Urges (QSU) [47]. This questionnaire is a brief version of the 32-item self-report questionnaire designed by Tiffany and Drobes [29] to measure smoking urges and cravings. We used the brief Spanish version of the questionnaire [48], which has been shown to have adequate psychometric properties and construct validity. This questionnaire consists of 10 items and is composed of two factors: (1) intentions and desires to smoke (QSU-Factor 1) and (2) the desire to smoke associated with the expectation of relief from negative affect (QSU-Factor 2). The reliability coefficients for both factors of the Spanish version of the questionnaire were α = 0.95 for Factor 1, and α = 0.93 for Factor 2 [48]. In our sample, the Cronbach’s alpha for the total scale was 0.88, being 0.87 for Factor 1 and 0.81 for Factor 2.The Perceived Stress Scale (PSS14) [49]. This self-report scale is made up of 14 items with a Likert-type response scale ranging from 0 to 4 to assess the degree to which a person appraises certain life situations as stressful. We used the Spanish adaptation of the scale [50], which has an internal consistency of α = 0.81 [51]. In our sample, the Cronbach’s alpha was 0.83.

Treatment outcome assessment:Abstinence at the end of treatment was self-reported and corroborated by assessing carbon monoxide (CO) in expired air using the Micro+ Smokerlyzer (Bedfont Scientific Ltd., Maidstone, Kent, UK). Participants were considered non-smokers if they reported no smoking for at least 24 h (not even one puff) at the end of treatment and had an expired CO ≤ 5 parts per million (ppm) [52]. Self-reported abstinence was checked to see if it matched the CO measurements for each participant. Those participants who self-reported abstinence but had CO measures of 6 ppm or higher were considered smokers. It is noteworthy that part of the sample was treated during the coronavirus disease 2019 (COVID-19) pandemic. Therefore, the abstinence measures of 55.38% of the participants were self-reported only. To test the possible bias that this might lead to, we compared abstinence percentages between those participants who had biochemical validation and those who were only self-reported. The results of the analysis revealed no statistically significant differences in abstinence rates (χ² = 1.708; *p* = 0.191).

### 2.3. Procedure

Participants were recruited from the community between January 2017 and June 2020 using posters in health centers and in different parts of the city, interviews in mass media, publications on the Smoking Cessation Unit’s social networks (Facebook and Instagram), referrals from primary care physicians or other health professionals, and other participants who had previously received the same treatment to quit smoking. Interested smokers contacted the unit by phone or email and were assigned an appointment for an individual pre-treatment assessment. A structured clinical assessment session was carried out in which sociodemographic characteristics and smoking-related variables were collected, and the above-mentioned questionnaires were completed. All participants provided written informed consent before the intervention.

All participants received the same multi-component cognitive behavioral smoking cessation treatment called “Programa para Dejar de Fumar” [53]. Two different delivery formats were used due to the COVID-19 pandemic: (1) in-person format in the Smoking Cessation and Addictive Disorders Unit during the pre-pandemic period from January 2017 to February 2020 and (2) through video calls during the pandemic period from March 2020 to June 2020. The treatment in both delivery formats consisted of eight 1 h group sessions once a week. Some of the treatment components included information about tobacco consequences, nicotine fading, smoking self-reporting, stimulus control, strategies for managing the withdrawal syndrome and relapse prevention (e.g., problem-solving training, weight control), and behavioral activation. Self-reported or biochemically verified smoking status was determined weekly during the treatment period (eight weeks). No pharmacotherapy was used. The participants did not receive financial compensation for their participation in the study.

### 2.4. Statistical Analysis

Descriptive statistics and Pearson correlations were calculated. Mediation analyses using PROCESS for SPSS, Version 4.0, were conducted to examine the indirect effect of perceived stress on abstinence outcomes at the end of smoking cessation treatment mediated through smoking urges. Model 4 was performed for both the simple and multiple mediation models. More specifically, a simple mediation model was conducted in which perceived stress was the independent variable (X), abstinence at the end of treatment was the dependent variable (Y; 0 = smoking; 1 = abstinence), and smoking urges (total QSU) was the mediator (M). A multiple mediation model was also performed in which the intentions and desires to smoke (QSU-Factor 1) and the desire to smoke associated with the expectation of relief from negative affect (QSU-Factor 2) were mediators in the relationship between perceived stress and abstinence at the end of treatment.

We examined differences related to sociodemographic data between the participants who received the different treatment delivery formats (in-person vs. video calls) and found significant differences in age (t = 2.397, *p* = 0.017), sex (χ^2^ = 4.861, *p* = 0.027), university or technical school (χ^2^ = 16.698, *p* = 0.001), and high school or general education diploma (χ^2^ = 7.687, *p* = 0.006). In order to control for possible confounding effects, these variables were included in the adjusted models. In addition, although no significant differences in abstinence rates were found between the treatment delivery formats, we included this variable in the model to account for its possible effects. All research variables were included in the model based on the scores obtained from the questionnaires described above.

Bootstrapping analysis with bias correction was used (10,000 resampling) to generate confidence intervals to analyze the study objectives [54,55]. In addition, reverse model analyses were performed to test QSU-total, QSU-Factor 1, and QSU-Factor 2 as predictor variables, as well as perceived stress as a mediator.

## 3. Results

### 3.1. Descriptive Analysis and Correlations Results

All participants in the study were Caucasian. Regarding smoking-related variables, participants smoked an average of 18.22 cigarettes per day (SE = 8.04), and 36.3% of the participants were cigarette dependents, according to FTCD scores (≥6). Descriptive data are presented in Table 1.

Pearson correlations are presented in Table 2. Perceived stress was significantly and positively correlated with all the proposed mediational variables (QSU-total, QSU-Factor 1, and QSU-Factor 2). Furthermore, QSU-total and QSU-Factor 2 were significantly and negatively correlated with the outcome variable (abstinence at the end of treatment).

### 3.2. Mediation Analyses Results

Regarding the simple mediation analysis, the full model was significant (*p* = 0.008; Nagelkerke R^2^= 0.098). The results showed a non-significant direct effect of PSS14 on abstinence at the end of treatment (β = −0.0154; SE = 0.0207, 95% BootCI [−0.0560, 0.0252]) (Figure 1 and Table 3). However, there was a significant indirect effect of PSS14 on abstinence through the QSU-total score in both the adjusted (β = −0.0164; SE = 0.0070, 95% BootCI [−0.0326, −0.0053]) and unadjusted models (β = −0.0152; SE = 0.0067, 95% BootCI [−0.0302, −0.0039]).

Regarding the multiple mediation analysis, the full model was significant (*p* = 0.006; Nagelkerke R^2^ = 0.110). This analysis with the two QSU factors (QSU-Factor 1 and QSU-Factor 2) showed non-significant direct effects of PSS14 on abstinence (β = −0.0107; SE = 0.0211, 95% BootCI [−0.0520, 0.0307]) (Figure 2 and Table 3). A significant indirect effect of PSS14 on abstinence at the end of treatment, through QSU-Factor 2 in both the adjusted (β = −0.0197; SE = 0.0093, 95% BootCI [−0.0415, −0.0050]) and unadjusted models (β = −0.0203; SE = 0.0093, 95% BootCI [−0.0412, −0.0051]) was found. However, no significant indirect effect was found through QSU-Factor 1 either in the adjusted (β = −0.0012; SE = 0.0052, 95% CI [−0.0123, 0.0092]) or unadjusted models (β = 0.0006; SE = 0.0056, 95% CI [−0.0105, 0.0120]). Finally, the total indirect effect was significant (β = −0.0209; SE = 0.0080, 95% CI [−0.0396, −0.0084]).

Concerning the reverse models analysis, in which PSS14 was a potential mediator in the relationships between QSU-total or QSU-Factors 1 and 2 and abstinence at the end of treatment, non-significant indirect effects were found when used as predictor variables: QSU-total (β = −0.0021; SE = 0.0032, 95% BootCI [−0.0086, 0.0041]), QSU-Factor 1 (β = −0.0036; SE = 0.0035, 95% BootCI [−0.0112, 0.0031]), and QSU-Factor 2 (β = −0.0035; SE = 0.0077, 95% BootCI [−0.0187, 0.0124]).

## 4. Discussion

This study aimed to investigate the mediating role of the urge to smoke in the relationship between perceived stress and abstinence at the end of a smoking cessation treatment program. Our results showed a significant indirect effect of perceived stress on abstinence through the total QSU score. This finding is consistent with previous studies that analyzed these variables separately, finding that higher perceived stress predicted stronger urges to smoke [42] and that urges to smoke were associated with smoking cessation treatment outcomes, such as lower abstinence rates at the end of smoking cessation treatment and higher relapse rates [35,37].

We did not obtain a significant direct effect of perceived stress on abstinence at the end of smoking cessation treatment. This finding aligns with previous studies, as Hooper and Kolar [27] and Manning et al. [28] found no association between perceived stress and abstinence at the end of treatment. Moreover, these authors did not find an association between perceived stress and smoking status at follow-ups. However, other studies have found a significant relationship between variables. Specifically, Slovinec et al. [25] reported that lower perceived stress increased the odds of being abstinent at follow-ups, and al’Absi et al. [26] found that higher perceived stress predicted relapse. One possible explanation for these conflicting findings may be the presence of mediating variables that were not assessed in these studies.

When the two dimensions of urges to smoke were analyzed as mediators, only the desire to smoke associated with the expectation of negative affect relief was significant (QSU-Factor 2). Despite the finding that the desire and intention to smoke related to the expectation of tobacco consumption as a pleasurable experience (QSU-Factor 1) has been associated with abstinence in previous studies [40], we did not find a significant effect as a mediator between perceived stress and abstinence at the end of smoking cessation treatment. Therefore, smokers with higher perceived stress might not smoke due to an expectation of pleasure but instead for the expectation of relief from negative affect. Previous studies analyzing the nonpharmacological determinants of smoking suggest that smoking behavior is related to positive pleasure seeking and to negative reinforcing effects—avoiding withdrawal symptoms [32]. According to this approach, we hypothesize that relief from the discomfort of the withdrawal syndrome could be generalized by the smoker to relief from the negative affect related to stress [56]. Therefore, smokers might learn to use cigarettes as a tool for managing stress symptoms [57].

Some limitations of this study should be mentioned. First, due to the COVID-19 pandemic, biochemical verification of abstinence was obtained for only half the sample. Expert recommendations [52,58] suggest that studies with limited in-person contact or data collected through the Internet or telephone, in which biochemical verification is not feasible, nevertheless provide reliable abstinence outcomes. Furthermore, we used treatment delivery format (in-person vs. online) as a covariate in the adjusted statistical analysis in order to account for the possible effects this could have. Second, the assessment questionnaires were self-reported, which could be influenced by biases such as social desirability. Third, perceived stress was measured at pre-treatment. Further studies are needed to analyze the influence of perceived stress assessed at other time points, such as post-treatment. Fourth, since the predictor and mediator variables were measured at the same moment, this study’s design does not allow us to make assumptions about the direction of the effects. For this reason, to gain confidence in the proposed models, different reverse models were analyzed, without finding significant effects in these alternative models. Finally, the present findings cannot be generalized to smokers from the general population because the sample was composed of treatment-seeking smokers.

This study also has some strengths. First, it has delved into the relationship between perceived stress and abstinence at the end of a smoking cessation treatment program. Therefore, this research contributes to filling a gap in the literature about the relationship between the variables studied. Second, this study was conducted with a large clinical sample. Finally, the analyses were carried out while controlling for numerous potential confounding effects.

The results of this study have important clinical implications, as they contribute to understanding the factors that may influence the effectiveness of smoking cessation treatments. For instance, including techniques for managing perceived stress in both daily activities and smoking-related situations (i.e., coping strategies training) could improve abstinence rates, as has been shown in previous studies [59]. Finally, although replication of the study findings is needed, our results could help clarify this association across different populations of adult smokers since those with mental health problems frequently mention the reduction of negative affect as one of the primary reasons for smoking [60,61,62].

## 5. Conclusions

In summary, this study highlights the relevance of smoking urges as mediators in the relationship between perceived stress and abstinence. Moreover, the analysis of the smoking urges dimensions emphasizes the relevance of the desire to smoke associated with the expectation of negative affect relief in a smoking cessation process in people with high perceived stress. Examining the mediator variables’ roles could contribute to tailoring interventions to treatment-seeking smokers’ characteristics and, consequently, improving treatment outcomes.

## Figures and Tables

**Figure 1 ijerph-20-01257-f001:**
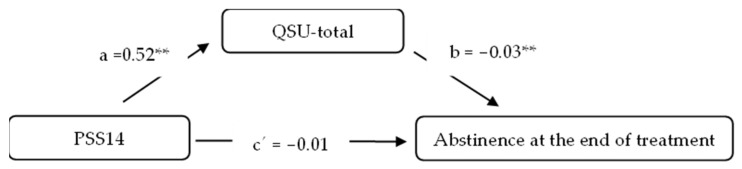
Simple mediation model with QSU-total as a mediator. *** p* < 0.01.

**Figure 2 ijerph-20-01257-f002:**
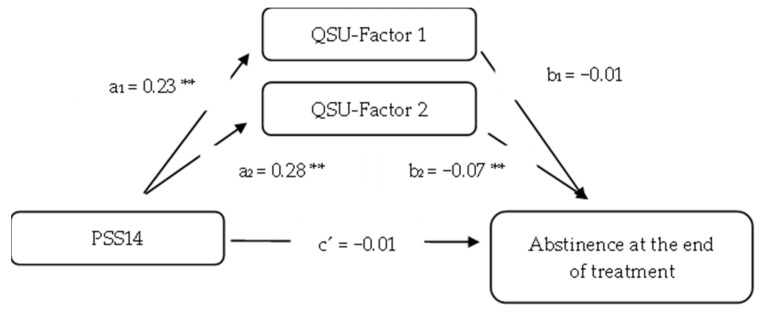
Multiple mediation model with QSU-Factors 1 and 2 as mediators. *** p* < 0.01.

**Table 1 ijerph-20-01257-t001:** Descriptive data of the study variables.

	*M* (*SD*)/% (*n*)
**Age (years)**	46.0 (11.1)
**Sex (female)**	58.5 (152)
**Education**	
**University or technical school**	50.8 (132)
**High school or general education diploma**	34.2 (89)
**Less than high school diploma**	14.6 (38)
**Without school diploma**	0.4 (1)
**Marital status**	
**Married**	45.8 (119)
**Single**	34.2 (89)
**Other marital status (e.g., divorced/separated, widowed)**	20.0 (52)
**FTCD ^1^**	4.8 (2.1)
**PSS14 ^2^**	23.8 (7.2)
**QSU-total ^3^**	31.9 (13.8)
**QSU-Factor 1**	20.7 (8.9)
**QSU-Factor 2**	11.2 (6.6)
**Abstinence at the end of treatment**	68.1 (177)

^1^ FTCD: Fagerström Test for Cigarette Dependence. ^2^ PSS14: Perceived Stress Scale. ^3^ QSU: Questionnaire of Smoking Urges.

**Table 2 ijerph-20-01257-t002:** Pearson correlations.

	1	2	3	4
**1 PSS14 ^1^**	-			
**2 QSU-total ^2^**	0.29 **	-		
**3 QSU-Factor 1**	0.22 **	0.92 **	-	
**4 QSU-Factor 2**	0.32 **	0.86 **	0.59 **	-
**5 Abstinence**	−0.08	−0.18 **	−0.12	−0.22 **

^1^ PSS14: Perceived Stress Scale. ^2^ QSU: Questionnaire of Smoking Urges. ** *p* < 0.01.

**Table 3 ijerph-20-01257-t003:** Results of simple and multiple mediation analyses controlling for covariates.

Simple Mediation						
		β	SE ^1^	*p*	LLCI ^2^	ULCI ^3^
	Direct					
PSS14 ^4^ → QSU-total ^5^		0.5160	0.1173	0.0000	0.2849	0.7471
QSU-total → Abstinence		−0.0317	0.0107	0.0031	−0.0528	−0.0107
PSS14 → Abstinence		−0.0154	0.0207	0.4568	−0.0560	0.0252
	Indirect					
PSS14 → QSU-total → Abstinence		−0.0164	0.0070		−0.0326	−0.0053
**Multiple Mediation**						
		**β**	**SE**	** *p* **	**LLCI**	**ULCI**
	Direct					
PSS14 → QSU-Factor 1		0.2322	0.0764	0.0026	0.0817	0.3828
PSS14 → QSU-Factor 2		0.2837	0.0556	0.0000	0.1742	0.3933
QSU-Factor 1 → Abstinence		−0.0051	0.0201	0.7976	−0.0444	0.0342
QSU-Factor 2 → Abstinence		−0.0693	0.0265	0.0089	−0.1212	−0.0174
PSS14 → Abstinence		−0.0107	0.0211	0.6134	−0.0520	0.0307
	Indirect					
PSS14 → QSU-Factor 1 → Abstinence		−0.0012	0.0052		−0.0123	0.0092
PSS14 → QSU-Factor 2 → Abstinence		−0.0197	0.0093		−0.0415	−0.0050

^1^ SE: Standard Error. ^2^ LLCI: lower limit confidence interval. ^3^ ULCI: upper limit confidence interval. ^4^ PSS14: Perceived Stress Scale. ^5^ QSU: Questionnaire of Smoking Urges.

## Data Availability

The data presented in this study are available upon request from the corresponding author.

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
