# Peer review of "Perceived Stress and Smoking Cessation: The Role of Smoking Urges"

_ijerph, 2023, doi:10.3390/ijerph20021257_

Round 1

Reviewer 1 Report

The article "Perceived stress and smoking cessation: the role of smoking urges" aims to explore the indirect effect of perceived stress on abstinence outcomes at the end of treatment through smoking urges and its dimensions in a Spanish seeking-treatment smokers sample.

The study is very interesting, and the manuscript is well written and was pleasant to read. For these reasons, I would like to congratulate the authors. However, I have some comments which are listed below:

Introduction

- Lines 27-32. Recent studies have shown that there is a relationship between smoking and suicide. Given its relevance, I would like you to name this aspect and how smoking cessation decreases its prevalence (Echeverria et al., 2021).

- Lines 47-49. Why do you think it occurs only in women?

- In D'Angelo, 2001 (ref. 18) higher stress is predictive of lower abstinence rates after treatment in women but not in men, and yet in Lawless, 2015 (ref. 15) higher stress is related to more smoking urge and more negative affect relief expectations in men but not in women. Hypothesize about this phenomenon.

Material and methods

- Why did you choose 6 cigarettes as the cutoff point for inclusion in the study?

- How did you ensure that the sample was free of mental health problems?

- Line 130-13: Could you specify the Crombach of the Spanish version? What are the reliability coefficients of both scales in the Spanish version?

- Why are 55% of abstinence measures "self-reported due to pandemic" if you only used the online format the months between March and June 2020?

- Were there differences in abstinence rates between before and during the pandemic?

- How long did it take from the start of therapy until abstinence was verified with self-report and Micro+ Smokerlyzer methos? Add this in procedure section

- How much time elapsed from the start of therapy until abstinence was verified with self-report and Micro+ Smokerlyzer methods? Add this in the procedure section

Results

- 203-205: You can omit this paragraph.

- I think it would be more appropriate to perform a chi-square between the sociodemographic variables of those who managed to quit smoking and those who did not, and then include in the regressions the variables with significant differences.

- Line 279.  If you are talking about a direct effect (line 271), then the variable should be a moderator

Reference:

Echeverria, I., Cotaina, M., Jovani, A., Mora, R., Haro, G., & Benito, A. (2021). Proposal for the Inclusion of Tobacco Use in Suicide Risk Scales: Results of a Meta-Analysis. International journal of environmental research and public health18(11), 6103. https://doi.org/10.3390/ijerph18116103

Author Response

The article "Perceived stress and smoking cessation: the role of smoking urges" aims to explore the indirect effect of perceived stress on abstinence outcomes at the end of treatment through smoking urges and its dimensions in a Spanish seeking-treatment smokers sample.

The study is very interesting, and the manuscript is well written and was pleasant to read. For these reasons, I would like to congratulate the authors. However, I have some comments which are listed below:

We thank the Reviewer for the comments on the study, which have helped us to improve the manuscript.

Introduction

- Lines 27-32. Recent studies have shown that there is a relationship between smoking and suicide. Given its relevance, I would like you to name this aspect and how smoking cessation decreases its prevalence (Echeverria et al., 2021).

Thank you for the suggestion. Following the reviewer recommendation, we have added this reference in the introduction section.

The following sentence has been added to the text:

Page 1, paraghrap 2, line 33-38:

- “However, research has shown that smoking cessation could reverse, at least in part, these tobacco-related negative effects [5–7]. For example, quitting smoking is related to improvements on mental health (Taylor et al., 2014). According to Hajek et al. (2010) and Streck et al. (2021), abstinence is associated with lower levels of stress and anxiety. Additionally, Kim et al. (2019) found that successful quitters have lower odds of stress than those who fail to quit. Other studies have shown that quitting smoking is associated with reductions in depressive symptoms (Martinez-Vispo et al., 2019, Rodriguez Cano et al., 2016), and even that could decrease suicide risk (Echevarria et al., 2021).”

- Lines 47-49. Why do you think it occurs only in women?

Thank you for this interesting question. Regarding the study showing that higher baseline perceived stress predicts lower abstinence rates in women but not in men after receiving a smoking cessation treatment (D´Ángelo, 2001), previous literature supports that stress, negative affect, and depression are important determinants of tobacco outcomes for women (Borelli et al., 1996; McKee et al., 2003; McKee et al., 2005). Moreover, research has shown that women could use smoking to cope with negative emotional states (Li et al., 2015; Xu et al., 2008; Weinberger et al., 2012).

- In D'Angelo, 2001 (ref. 18) higher stress is predictive of lower abstinence rates after treatment in women but not in men, and yet in Lawless, 2015 (ref. 15) higher stress is related to more smoking urge and more negative affect relief expectations in men but not in women. Hypothesize about this phenomenon.

Thank you to the reviewer for this comment that allows us to clarify this issue. These studies use different dependent variables: abstinence vs. smoking urges. In the article of D'Angelo et al. (2021), the authors examied gender differences in different variables (for example, perceived stress) as predictors of 1 year of smoking abstinence. However, Lawless et al. (2015) investigated the association between perceived stress, nicotine-related symptomatology (nicotine withdrawal, nicotine dependence, and smoking urges), sociodemographic variables (e.g., income), and smoking behavior (number of cigarettes smoked per day) according to sex. Therefore, abstinence was not analysed in this study.

Material and methods

- Why did you choose 6 cigarettes as the cutoff point for inclusion in the study?

We appreciate the reviewer comment. No consensus exists regarding the definition of a daily smoker. For instance, studies as Lou et al. (2013) used the criteria of smoking 1 or more cigarettes per day for the last year, Ward et al. (2013) of smoking 5 cigarettes or more per day, Scherphof et al. (2014) smoking at least 7 cigarettes/day and Tuisku et al. (2016) for smoking at least 100 cigarettes in their lifetime and have smoked daily for the past month. Therefore, there is great heterogeneity in the definition of a daily smoker for inclusion criteria in smoking cessation treatment studies.

The inclusion criteria of six cigarettes smoked per day used in our study is more restrictive than the used in most of the aforementioned studies in the previous paragraph. We established the cut-off point based on the smokers profile attending to our Unit and the review of previous studies with seeking treatment smokers (Becoña et al., 2014).

- How did you ensure that the sample was free of mental health problems?

Thank you for the question. In the present study, the presence of any mental health problem was not an exclusion criterion. We only excluded participants if they reported a diagnosis of a severe mental disorder (schizophrenia or bipolar disorder). During the baseline assessment, participants are asked if they have previously received any of these diagnoses. Therefore, only those reporting a diagnosis of a severe mental disorder were excluded from the study.

- Line 130-13: Could you specify the Crombach of the Spanish version? What are the reliability coefficients of both scales in the Spanish version?

We appreciate this suggestion. Following the reviewer recommendation, we have added the Cronbach alpha of the Spanish version in both scales. The following sentences were added:

Page 3, paragraph 8, lines 144-147:

- “Brief version of the Questionnaire of Smoking Urges (QSU) [47]. This questionnaire is a brief version of the 32-item self-report questionnaire designed by Tiffany and Drobes [29] to measure smoking urges and cravings. We used the brief Spanish version of the questionnaire [48], which has been shown to have adequate psy-chometric properties and construct validity. This questionnaire consists of 10 items and is composed of two factors: (1) intentions and desires to smoke (QSU-Factor 1) and (2) the desire to smoke associated with the expectation of relief from negative affect (QSU-Factor 2). Furthermore, reliability coefficients for both factors of the Spanish version of the questionnaire were α = .95 for Factor 1,  and α = .93 for Factor 2 (Cepeda-Benito & Reig-Ferrer, 2004). In our sample, the Cronbach alpha for the total scale was .88, being .87 for Factor 1 and .81 for Factor 2

Page 4, paragraph 1, lines 153-154:

- “Perceived Stress Scale (PSS14) [49]. This self-report scale is made up of 14-items with a Likert-type response scale ranging from 0 to 4 to assess the degree to which a person appraises certain life situations as stressful. We used the Spanish adaptation of the scale [50], which has an internal consistency of α = .81 [51]. In our sample the Cronbach alpha was .83.

- Why are 55% of abstinence measures "self-reported due to pandemic" if you only used the online format the months between March and June 2020?

Thank you for the question. The 44.62% of the sample was collected between January 2017 and March 2020, while the 55.38% was collected between March and June 2020. The online format was implemented in March 2020 due to the COVID-19 social contact restrictions. When we began to carry out the treatment in online format, we experienced a large increase of seeking treatment smokers who wanted to quit. This could be due to smokers' concern for their health increased during the pandemic, given the relationship between tobacco use, greater severity of COVID-19 symptoms, and a higher mortality risk (Engin et al., 2020; OMS, 2020), which may have influenced the increased motivation for quitting.

Additionally, the online format allows to remove some of the barriers associated with in-person treatments. For example, this delivery format facilitates reaching smokers unable to travel to receive treatment or living too far to attend in person. The use of the online format is supported by previous literature showing that the use of video-calls for smoking cessation treatment has some advantages compared to traditional in-person treatments, such as greater recruitment and more demographically representative samples (Dahne et al., 2020).

- Were there differences in abstinence rates between before and during the pandemic?

We appreciate this comment. In order to control for potential confounders, possible differences in abstinence rates between participants who received treatment before the pandemic (in person format) and during the pandemic (video calls) were analysed. No significant differences were found between both groups of participants. This phrase is specified in the statistical analysis section, in page 5, paragraph 2 and lines 211-213: “In addition, although no significant differences in abstinence rates were found between both treatment delivery formats, we included this variable in the model to account for its possible effects”.

- How long did it take from the start of therapy until abstinence was verified with self-report and Micro+ Smokerlyzer methos? Add this in procedure section

Thank you for your comment. We have clarified this information in the procedure section.

The following sentence was included:

Page 4, paragraph 4, lines 189-190:

-  “The treatment in both delivery formats consisted of eight 1-hour group sessions once a week. Some of the treatment components are information about tobacco consequences, nicotine fading, smoking self-report, stimulus control, strategies for managing withdrawal syndrome and relapse-prevention (e.g., problem-solving training, weight control), and behavioral activation. Self-reported or biochemically verified smoking status was conducted weekly during the treatment period (eight weeks). No pharmacotherapy was used, and participants did not receive economic compensation for their participation in the study.”

Results

- 203-205: You can omit this paragraph.

Thank you for the suggestion. Following the reviewer comment, we have removed that paragraph from the results section (page 5, paragraph 4, lines 221-223).

- I think it would be more appropriate to perform a chi-square between the sociodemographic variables of those who managed to quit smoking and those who did not, and then include in the regressions the variables with significant differences.

We thank the reviewer for the comment. As suggested, we examined the differences related to sociodemographic variables between the participants who quit smoking and those who did not, founding no significant differences: age (t = 1.425, p = .155), sex (χ2 = 1.491, p = .222), education (χ2 = 3.860, p = .277) and marital status (χ2 = 0.650, p = .722). Given the non-significant differences based on smoking status, these variables were not included in the statistical analyses as covariates.

- Line 279.  If you are talking about a direct effect (line 271), then the variable should be a moderator

We thank the reviewer for allowing us to clarify this issue. According to Hayes & Little (2018) mediation is based on how the effect of an antecedent variable X on a final consequent variable Y can be influenced in two ways: direct and indirectly. The direct effect is the influence of X on Y without taking into account the mediating variable (M) and the indirect effect is the effect of X on Y through M. Therefore, when discussing the mixed findings of previous literature (page 8, paragraph 2, lines 297-306) we highlight that this inconsistency could be due to the influence of mediating variables.

Reference

Echeverria, I., Cotaina, M., Jovani, A., Mora, R., Haro, G., & Benito, A. (2021). Proposal for the Inclusion of Tobacco Use in Suicide Risk Scales: Results of a Meta-Analysis. International journal of environmental research and public health18(11), 6103. https://doi.org/10.3390/ijerph18116103

We have included this reference in the manuscript as suggested.

References

Becoña, E., López-Durán, A., Fernández del Río, E., & Martínez, Ú. (2014). Changes in the profiles of smokers seeking cessation treatment and in its effectiveness in Galicia (Spain) 2001-10. BMC Public Health14, 613. https://doi.org/10.1186/1471-2458-14-613

Borrelli, B., Bock, B., King, T., Pinto, B., & Marcus, B. H. (1996). The impact of depression on smoking cessation in women. American Journal of Preventive Medicine12(5), 378–387.

Cepeda-Benito, A., & Reig-Ferrer, A. (2004). Development of a Brief Questionnaire of Smoking Urges—Spanish. Psychological Assessment, 16(4), 402–407. https://doi.org/10.1037/1040-3590.16.4.402

Dahne, J., Tomko, R. L., McClure, E. A., Obeid, J. S., & Carpenter, M. J. (2020). Remote Methods for Conducting Tobacco-Focused Clinical Trials. Nicotine & Tobacco Research22(12), 2134–2140. https://doi.org/10.1093/ntr/ntaa105

Engin, A. B., Engin, E. D., & Engin, A. (2020). Two important controversial risk factors in SARS-CoV-2 infection: Obesity and smoking. Environmental Toxicology and Pharmacology, 78, 103411. https://doi.org/10.1016/j.etap.2020.103411 

Hajek, P., Taylor, T., & McRobbie, H. (2010). The effect of stopping smoking on perceived stress levels. Addiction105(8), 1466–1471. https://doi.org/10.1111/j.1360-0443.2010.02979.x

Hayes, A. F., & Little, T. D. (2018). Introduction to mediation, moderation, and conditional process analysis: a regression-based approach. Guilford Press.

Kim, S. J., Chae, W., Park, W. H., Park, M. H., Park, E. C., & Jang, S. I. (2019). The impact of smoking cessation attempts on stress levels. BMC Public Health19(1), 267. https://doi.org/10.1186/s12889-019-6592-9

Li, H. C., Chan, S. S., & Lam, T. H. (2015). Smoking among Hong Kong Chinese women: behavior, attitudes and experience. BMC public health15, 183. https://doi.org/10.1186/s12889-015-1529-4

Lou, P., Zhu, Y., Chen, P., Zhang, P., Yu, J., Zhang, N., Chen, N., Zhang, L., Wu, H., & Zhao, J. (2013). Supporting smoking cessation in chronic obstructive pulmonary disease with behavioral intervention: a randomized controlled trial. BMC Family Practice14, 91. https://doi.org/10.1186/1471-2296-14-91

Martínez-Vispo, C., Rodríguez-Cano, R., López-Durán, A., Senra, C., Fernández del Río, E, &Becoña, E. (2019) Cognitive-behavioral treatment with behavioral activation for smoking cessation: Randomized controlled trial. PLoS ONE, 14(4), e0214252. https://doi.org/10.1371/journal.pone.0214252

McKee, S. A., Maciejewski, P. K., Falba, T., & Mazure, C. M. (2003). Sex differences in the effects of stressful life events on changes in smoking status. Addiction (Abingdon, England)98(6), 847–855. https://doi.org/10.1046/j.1360-0443.2003.00408.x

McKee, S. A., O'Malley, S. S., Salovey, P., Krishnan-Sarin, S., & Mazure, C. M. (2005). Perceived risks and benefits of smoking cessation: gender-specific predictors of motivation and treatment outcome. Addictive Behaviors30(3), 423–435. https://doi.org/10.1016/j.addbeh.2004.05.027

Organización Mundial de la Salud (OMS) (2020). Declaración de la OMS: consumo de tabaco y COVID-19. https://www.who.int/es/news/item/11-05-2020-who-statement-tobacco-use-and-covid-19 

Rodríguez-Cano, R., López-Durán, A., Fernández del Río, E. F., Martínez-Vispo, C., Martínez, Ú., & Becoña, E. (2016). Smoking cessation and depressive symptoms at 1-, 3-, 6-, and 12-months follow-up. Journal of Affective Disorders, 191, 94–99. https://doi.org/10.1016/j.jad.2015.11.042

Scherphof, C. S., van den Eijnden, R. J., Engels, R. C., & Vollebergh, W. A. (2014). Long-term efficacy of nicotine replacement therapy for smoking cessation in adolescents: a randomized controlled trial. Drug and Alcohol Dependence140, 217–220. https://doi.org/10.1016/j.drugalcdep.2014.04.007

Streck, J. M., Luberto, C. M., Muzikansky, A., Skurla, S., Ponzani, C. J., Perez, G. K., Hall, D. L., Gonzalez, A., Mahaffey, B., Rigotti, N. A., Ostroff, J. S., & Park, E. R. (2021). Examining the effects of stress and psychological distress on smoking abstinence in cancer patients. Preventive Medicine Reports23, 101402. https://doi.org/10.1016/j.pmedr.2021.101402

Taylor, G., McNeill, A., Girling, A., Farley, A., Lindson-Hawley, N. y Aveyard, P. (2014). Change in mental health after smoking cessation: systematic review and meta-analysis. The BMJ348, g1151. https://doi.org/10.1136/bmj.g1151

Tuisku, A., Salmela, M., Nieminen, P., & Toljamo, T. (2016). Varenicline and nicotine
patch therapies in young adults motivated to quit smoking: A randomized, placebocontrolled, prospective study. Basic & Clinical Pharmacology & Toxicology, 119(1), 78–84.

Xu, J., Azizian, A., Monterosso, J., Domier, C. P., Brody, A. L., Fong, T. W., & London, E. D. (2008). Gender effects on mood and cigarette craving during early abstinence and resumption of smoking. Nicotine & Tobacco Research10(11), 1653–1661. https://doi.org/10.1080/14622200802412929

Ward, K. D., Asfar, T., Al Ali, R., Rastam, S., Weg, M. W., Eissenberg, T., & Maziak, W. (2013). Randomized trial of the effectiveness of combined behavioral/pharmacological smoking cessation treatment in Syrian primary care clinics. Addiction108(2), 394–403. https://doi.org/10.1111/j.1360-0443.2012.04048.x

Weinberger, A. H., & McKee, S. A. (2012). Gender differences in smoking following an implicit mood induction. Nicotine & Tobacco Research14(5), 621–625. https://doi.org/10.1093/ntr/ntr198

Reviewer 2 Report

Thanks to the authors for sharing their interesting research. I would like to share with the authors some comments:

·         The authors provide a good theoretical review, but it lacks research highlighting some important aspects of quitting smoking (e.g., related to health conditions). I recommend expanding the theoretical review a bit; the authors can draw on the following or some other sources:

Kim, SJ., Chae, W., Park, WH. et al. (2019). The impact of smoking cessation attempts on stress levels. BMC Public Health 19, 267. https://doi.org/10.1186/s12889-019-6592-9

Hajek, P., Taylor, T., McRobbie, H. (2010). The effect of stopping smoking on perceived stress levels. Addiction 105(8), 1466-71. https://doi.org/10.1111/j.1360-0443.2010.02979.x

Streck, JM., Luberto, CM., Muzikansky, A., et al. (2021).  Examining the effects of stress and psychological distress on smoking abstinence in cancer patients. Preventive Medicine Reports 23, 101402. https://doi.org/10.1016/j.pmedr.2021.101402

·         The authors describe measures, including the fact that they used adapted Spanish versions. I think it would have been better if the authors had also specified Cronbach's alpha for their sample, especially since the Fagerström Test for Cigarette Dependence has questionable internal reliability, and the authors infer cigarette addiction from it.

·         It is not known whether the authors calculated the sample size before conducting the study. This information should be added to the Method.

Author Response

Thanks to the authors for sharing their interesting research. I would like to share with the authors some comments:

We appreciate all the Reviewer comments and suggestions about the present study, these will help us to improve the manuscript.

- The authors provide a good theoretical review, but it lacks research highlighting some important aspects of quitting smoking (e.g., related to health conditions). I recommend expanding the theoretical review a bit; the authors can draw on the following or some other sources:

Kim, SJ., Chae, W., Park, WH. et al. (2019). The impact of smoking cessation attempts on stress levels. BMC Public Health 19, 267. https://doi.org/10.1186/s12889-019-6592-9

Hajek, P., Taylor, T., McRobbie, H. (2010). The effect of stopping smoking on perceived stress levels. Addiction 105(8), 1466-71. https://doi.org/10.1111/j.1360-0443.2010.02979.x

Streck, JM., Luberto, CM., Muzikansky, A., et al. (2021).  Examining the effects of stress and psychological distress on smoking abstinence in cancer patients. Preventive Medicine Reports 23, 101402. https://doi.org/10.1016/j.pmedr.2021.101402

We appreciate this comment. Following the reviewer suggestion, we have added some important aspects of quitting smoking based on the mentioned references. We have added the following phrases in the introduction section:

Page 1, paraghrap 2, line 33-38:

“…Furthermore, smoking contributes to physical illnesses, such as cancer, cardiovascular and respiratory diseases [3], and mental health problems like depression, anxiety, or schizophrenia [4]. However, research has shown that smoking cessation could reverse, at least in part, these tobacco-related negative effects [5–7]. For example, quitting smoking is related to improvements on mental health (Taylor et al., 2014). According to Hajek et al. (2010) and Streck et al. (2021), abstinence is associated with lower levels of stress and anxiety. Additionally, Kim et al. (2019) found that successful quitters have lower odds of stress than those who fail to quit. Other studies have shown that quitting smoking is associated with reductions in depressive symptoms (Martinez-Vispo et al., 2019, Rodriguez Cano et al., 2016), and even that could decrease suicide risk (Echevarria et al., 2021).

- The authors describe measures, including the fact that they used adapted Spanish versions. I think it would have been better if the authors had also specified Cronbach's alpha for their sample, especially since the Fagerström Test for Cigarette Dependence has questionable internal reliability, and the authors infer cigarette addiction from it.

Thank you for your suggestion. As recommended, we have calculated the Cronbach's alpha of each instrument. In our sample, Cronbach alpha for the Fagerström Test for Cigarette Dependence was .59, for the Questionnaire of Smoking Urges (QSU) was .88, being .87 for the Factor 1 and .81 for Factor 2, and for the Perceived Stress Scale (PSS14) was .83. The reliability of the Fagerström Test for Cigarette Dependence in our sample is similar to the reported in other studies as Meneses-Gaya et al., (2009) who found a Cronbach alpha range of .55-.74 for this instrument in different studies. However, if it is of note that this measure is only used for descriptive purposes of the sample.

The information added in the manuscript is as follow:

  • Page 3, paragraph 7, lines 133-135:

“Fagerström Test for Cigarette Dependence (FTCD) [44]: This self-report questionnaire presents 6 items with 2 to 4 response alternatives. The cut-off point that indicates high nicotine dependence is a score ≥ 6 [45]. We used the Spanish adaptation of the scale, which has a Cronbach's alpha coefficient of .66 [46]. In our sample the Cronbach alpha was .59

  • Page 3, paragraph 8, lines 144-147:

“Brief version of the Questionnaire of Smoking Urges (QSU) [47]. This questionnaire is a brief version of the 32-item self-report questionnaire designed by Tiffany and Drobes [29] to measure smoking urges and cravings. We used the brief Spanish version of the questionnaire [48], which has been shown to have adequate psychometric properties and construct validity. This questionnaire consists of 10 items and is composed of two factors: (1) intentions and desires to smoke (QSU-Factor 1) and (2) the desire to smoke associated with the expectation of relief from negative affect (QSU-Factor 2). Furthermore, reliability coefficients for both factors of the Spanish version of the questionnaire were α = .95 for Factor 1,  and α = .93 for Factor 2 (Cepeda-Benito & Reig-Ferrer, 2004). In our sample, the Cronbach alpha for the total scale was .88, being .87 for Factor 1 and .81 for Factor 2”.

  • Page 4, paragraph 1, lines 153-154:

“Perceived Stress Scale (PSS14) [49]. This self-report scale is made up of 14-items with a Likert-type response scale ranging from 0 to 4 to assess the degree to which a person appraises certain life situations as stressful. We used the Spanish adaptation of the scale [50], which has an internal consistency of α = .81 [51]. In our sample the Cronbach alpha was .83.

- It is not known whether the authors calculated the sample size before conducting the study. This information should be added to the Method.

We appreciate the reviewer comment. Due to the descriptive nature of the present study, we did not calculate the sample size a priori. However, due to the clinical nature of the present sample (260 seeking treatment smokers), the sample size could be considered adequate.

References

Cepeda-Benito, A., & Reig-Ferrer, A. (2004). Development of a Brief Questionnaire of Smoking Urges—Spanish. Psychological Assessment, 16(4), 402–407. https://doi.org/10.1037/1040-3590.16.4.402

Echeverria, I., Cotaina, M., Jovani, A., Mora, R., Haro, G., & Benito, A. (2021). Proposal for the Inclusion of Tobacco Use in Suicide Risk Scales: Results of a Meta-Analysis. International journal of environmental research and public health18(11), 6103. https://doi.org/10.3390/ijerph18116103

Martínez-Vispo, C., Rodríguez-Cano, R., López-Durán, A., Senra, C., Fernández del Río, E, &Becoña, E. (2019) Cognitive-behavioral treatment with behavioral activation for smoking cessation: Randomized controlled trial. PLoS ONE, 14(4), e0214252. https://doi.org/10.1371/journal.pone.0214252

Meneses-Gaya, I. C., Zuardi, A. W., Loureiro, S. R., & Crippa, J. A. (2009). Psychometric properties of the Fagerström Test for Nicotine Dependence. Jornal Brasileiro de Pneumologia35(1), 73–82. https://doi.org/10.1590/s1806-37132009000100011

Rodríguez-Cano, R., López-Durán, A., Fernández del Río, E. F., Martínez-Vispo, C., Martínez, Ú., & Becoña, E. (2016). Smoking cessation and depressive symptoms at 1-, 3-, 6-, and 12-months follow-up. Journal of Affective Disorders, 191, 94–99. https://doi.org/10.1016/j.jad.2015.11.042

Taylor, G., McNeill, A., Girling, A., Farley, A., Lindson-Hawley, N. y Aveyard, P. (2014). Change in mental health after smoking cessation: systematic review and meta-analysis. The BMJ348, g1151. https://doi.org/10.1136/bmj.g1151

Reviewer 3 Report

Thank you for the opportunity to review this study entitled “Perceived stress and smoking cessation: the role of smoking urges.” (ijerph-2118276).

The study shows a mediation model to explore the indirect effect of perceived stress on abstinence at the end of treatment through smoking urges. The research involved a sample of 260 seeking-treatment smokers who received a psychological smoking cessation treatment.

In my opinion, the research topic is relevant, and the study is interesting. Parallelly, there are some issues that need to be addressed before the paper will be suitable for publication.

·       Abstract: the information about the sample should be deepened (Mean age and SD? Percentage of men and women?) to provide a clear picture of what will be presented in the paper.

·       Statistical analysis: please, detail the model implementation by adding the model number according to Hayes (2018). I think that Model 4 and Model 6 have been used.

·       My major concern regards the TOTAL effect, which was not reported. I am afraid it is not significant given the lack of significance in the correlation between PSS and Abstinence. In this case it would not make sense to perform mediation. Please report the total effect.

·       The study is cross-sectional. This should be clarified by the authors within the limitations of the study. In light of this, the relationships examined in the model must have plausible directions according to the scientific literature and must be theoretically grounded. Therefore, it makes no sense for the authors to elaborate "reverse model analysis" if they do not offer the theoretical and scientific bases to support it.

Best wishes

Author Response

Thank you for the opportunity to review this study entitled “Perceived stress and smoking cessation: the role of smoking urges.” (ijerph-2118276).

The study shows a mediation model to explore the indirect effect of perceived stress on abstinence at the end of treatment through smoking urges. The research involved a sample of 260 seeking-treatment smokers who received a psychological smoking cessation treatment.

In my opinion, the research topic is relevant, and the study is interesting. Parallelly, there are some issues that need to be addressed before the paper will be suitable for publication.

We thank the Reviewer for the comments about the present study. The suggestions provided have helped us to improve the quality of the manuscript. Please, find our point-by-point answers as follow.

  • Abstract: the information about the sample should be deepened (Mean age and SD? Percentage of men and women?) to provide a clear picture of what will be presented in the paper

We appreciate this comment. Following the reviewer suggestion, we have added this information in the abstract.

Page 1, paragraph 1, lines 12-13:

“Despite that perceived stress is related to abstinence smoking outcomes, no studies have investigated the mediational effect of specific tobacco-related variables on this relationship. The study aims to explore the indirect effect of perceived stress on abstinence at the end of treatment through smoking urges. The sample comprised 260 seeking-treatment smokers (58.5% female; Mage= 46.00; SD= 11.1) who received a psychological smoking cessation treatment.”

  • Statistical analysis: please, detail the model implementation by adding the model number according to Hayes (2018). I think that Model 4 and Model 6 have been used.

Thank you for your comment, since it allows us to clarify this question. We used model 4 of the PROCESS for SPSS, which analyzes the influence of a predictor variable (X) on the outcome variable (Y) through a mediator(s) variable(s) (M). This model can be used for simple mediation models (one mediator variable) and parallel multiple mediator models (more than one mediator variable). In model 4, in contrast to other models, the order of the mediator variables introduced is not consequential (Hayes & Little, 2018).  

We did not conduct model 6 since it is used for multiple serial mediations where a causal sequence is taken into account when examining the effects of mediators on the outcome: “the order of the variables in the list of mediators is taken literally as the causal sequence, with the first mediator variable in the list causally prior to the second in the list, and so forth” (Hayes & Little, 2018).

To clarify this question, we have added the following information to the manuscript:

Page 4-5, paragraph 5, lines 197-201:

“Descriptive statistics and Pearson correlations were calculated for the present study. Mediation analyses using PROCESS for SPSS, Version 4.0, were conducted to examine the indirect effect of perceived stress on abstinence outcomes at the end of a smoking cessation treatment mediated through smoking urges. Model 4 was performed for both, simple and multiple mediation models. More specifically, a simple mediation model was conducted in which perceived stress was the independent variable (X), abstinence at the end of treatment was the dependent variable (Y; 0 = smoking; 1 = abstinence), and smoking urges (total QSU) was the mediator (M). Furthermore, a multiple mediation model was also performed, in which the intentions and desires to smoke (QSU-Factor 1) and the desire to smoke associated with the expectation of relief from negative affect (QSU-Factor 2) were mediators in the relationship between perceived stress and abstinence at the end of treatment.”

  • My major concern regards the TOTAL effect, which was not reported. I am afraid it is not significant given the lack of significance in the correlation between PSS and Abstinence. In this case it would not make sense to perform mediation. Please report the total effect.

We appreciate this comment since this question could be confusing. Mediation models performed with PROCESS for SPSS do not report the total effect (c) when a dichotomous variable is used as the outcome variable (Hayes & Little, 2018). Regarding the question raised by the reviewer, in the present study, the outcome variable used is dichotomous (abstinence vs. smoking). Therefore, the total effect is not calculated by PROCESS. However, we have included the information regarding the full model signification in the manuscript. Specifically, the full model p value for the simple mediation model was .008; (Nagelkerke R2= .098); and for the multiple mediation model was .006 (Nagelkerke R2= .110).

Page 6, paragraph 2, lines 243-244:

“Regarding the simple mediation analysis, the full model was significant (p = .008; Nagelkerke R2= .098). Results showed a non-significant direct effect of PSS14 on abstinence at the end of treatment (β = -0.0154; SE = 0.0207, 95% BootCI [-0.0560, 0.0252]) (Figure 1 and Table 3). However, there was a significant indirect effect of PSS14 on abstinence through the QSU-total score in both adjusted (β = -0.0164; SE = 0.0070, 95% BootCI [-0.0326, -0.0053]) and unadjusted models (β = -0.0152; SE = 0.0067, 95% BootCI [-0.0302, -0.0039]).”

 Page 6, paragraph 3, lines 252-253:

Regarding the multiple mediation analysis, the full model was significant (p = .006; Nagelkerke R2= .110). This analysis with the two QSU factors (QSU-Factor 1 and QSU-Factor 2) showed non-significant direct effects of PSS14 on abstinence (β = -0.0107; SE = 0.0211, 95% BootCI [-0.0520, 0.0307]) (Figure 2 and Table 3). A significant indirect effect of PSS14 on abstinence at the end of treatment, through QSU-Factor 2 in both adjusted (β = -0.0197; SE = 0.0093, 95% BootCI [-0.0415, -0.0050]) and unadjusted models (β = -0.0203; SE = 0.0093, 95% BootCI [-0.0412, -0.0051]) was found. However, no significant indirect effect was found through QSU-Factor 1 either in the adjusted (β = -0.0012; SE = 0.0052, 95% CI [-0.0123, 0.0092]) or the unadjusted model (β = 0.0006; SE = 0.0056, 95% CI [-0.0105, 0.0120]). Finally, the total indirect effect was significant (β = -0.0209; SE = 0.0080, 95% CI [-0.0396, -0.0084])”

  • The study is cross-sectional. This should be clarified by the authors within the limitations of the study. In light of this, the relationships examined in the model must have plausible directions according to the scientific literature and must be theoretically grounded. Therefore, it makes no sense for the authors to elaborate "reverse model analysis" if they do not offer the theoretical and scientific bases to support it.

We appreciate this suggestion. As the reviewer points out, the predictor and mediator variables are measured at the same time point, which prevents causal inferences in the direction of the effects. However, we have conducted the reverse models in order to gain confidence in our results. We have included this information in the limitations section to clarify this question.

Page 8, paragraph 4, lines 331-334:

“…Thirdly, perceived stress was measured at pre-treatment. Further studies are needed to analyze the influence of perceived stress assessed at other time points, such as posttreatment. Fourthly, since the predictor and mediator variables were measured in the same moment, the design of the present study does not allow to make assumptions about the direction of the effects. For this reason, to gain confidence in the proposed models, different reverse models were conducted, not founding significant effects in these alternative models. Finally, the present findings cannot be generalized to smokers from the general population because the sample was composed of treatment-seeking smokers”

Reference

 Hayes, A. F., & Little, T. D. (2018). Introduction to mediation, moderation, and conditional process analysis: a regression-based approach. Guilford Press.

Round 2

Reviewer 1 Report

The authors have answered all my questions. Thank you.

Reviewer 3 Report

The authors resolved all the issues I raised.

Best wishes